# Assessing population exposure to coastal flooding due to sea level rise

Mathew E. Hauer [1,2✉], Dean Hardy [3,4], Scott A. Kulp[5], Valerie Mueller [6,7], David J. Wrathall[8] & Peter U. Clark[8,9]

The exposure of populations to sea-level rise (SLR) is a leading indicator assessing the impact of future climate change on coastal regions. SLR exposes coastal populations to a spectrum of impacts with broad spatial and temporal heterogeneity, but exposure assessments often narrowly define the spatial zone of flooding. Here we show how choice of zone results in differential exposure estimates across space and time. Further, we apply a spatio-temporal flood-modeling approach that integrates across these spatial zones to assess the annual probability of population exposure. We apply our model to the coastal United States to demonstrate a more robust assessment of population exposure to flooding from SLR in any given year. Our results suggest that more explicit decisions regarding spatial zone (and associated temporal implication) will improve adaptation planning and policies by indicating the relative chance and magnitude of coastal populations to be affected by future SLR.

[1] Department of Sociology, Florida State University, Tallahassee, FL 32306, USA. [2] Center for Demography and Population Health, Florida State University, Tallahassee, FL 32306, USA. [3] School of the Earth, Ocean, & Environment, University of South Carolina, Columbia, SC 29208, USA. [4] Department of Geography, University of South Carolina, Columbia, SC 29208, USA. [5] Climate Central, Princeton, NJ 08542, USA. [6] School of Politics and Global Studies, Arizona State University, Tempe, AZ 85287-3902, USA. [7] International Food Policy Research Institute, Washington, DC 20005, USA. [8] College of Earth, Ocean, and Atmospheric Sciences, Oregon State University, Corvallis, OR 97331-5503, USA. [9] School of Geography and Environmental Sciences, University of Ulster, Coleraine, Northern Ireland BT52 1SA, UK. ✉email: mehauer@fsu.edu

Sea-level rise (SLR) accompanying climate change will cause significant and costly impacts in the 21st century and beyond. Avoiding adverse consequences depends on our ability to undertake accurate assessments of the populations already affected, and those projected to be affected, to inform adaptation planning and the ability to adapt to such consequences[1]. Research has long sought to identify the impact of SLR on coastal areas[2–4], with an increasing focus on estimating exposed populations and associated assets[2,5,6]. The human population is concentrated in the low-elevation coastal zone (LECZ; those <10 m above sea level) with more than 600 million people living in the LECZ globally[7], and despite the increasing rate of SLR related flooding[8], the global LECZ population is growing: more than 1 billion are forecast to live in the coastal zone by 2060[5].

Scientific assessments estimating the populations affected by SLR date back at least four decades and are relatively common in the scientific literature. Intense interest in this topic is due to the magnitude and severity of SLR flooding as a climate impact, the clear potential implications for human migration[9], the growing size of global coastal populations[5], the relative simplicity of producing estimates[3], and the increasing availability of both geophysical and population data products from local to global scales[7]. However, the magnitude of the population estimated to be affected ranges widely across studies. At the global level, the population estimated to be affected by SLR ranges from a low of 88 million[10] to a high of 1.4 billion[5]. These wide-ranging estimates can be attributed to several considerations, including: (1) differing spatial zones of "at-risk" that influence estimates of how many people will be affected by SLR, and (2) differing temporal horizons implied by any given spatial zone that affect estimates of when increased flooding and associated impacts due to SLR will occur. A third major consideration is the deployment of different datasets and methods to calculate exposure. Examining the contribution of different datasets and methods to wide-ranging estimates of SLR exposure is beyond the scope of this paper. For simplicity, we use the term "spatial zone" throughout the paper to describe inland areas relative to the coastline. However, we recognize that in some cases for these zones, we are discussing areas that are representative of "spatio-temporal zones" such as the 100-year flood plain, which while explicitly spatial, is dependent upon the temporal notion of a 1% chance of flooding in any given year.

The modeling choices around spatial zone of population affected by SLR imply a temporal horizon for when impacts will unfold. These temporal horizons can be forecast from as short as 100 years[11] to as long as 2000 years or longer[6,12], pushing SLR impacts deep into the future. However, permanent inundation is not the most immediate impact of SLR. Regular daily to annual tidal flooding (e.g., nuisance flooding) events are likely to be the most disruptive to life in the near term[8], and related impacts are already occurring in many parts of the world (e.g. coastal erosion[13], coastal flooding[14], and saltwater intrusion[15]). Yet each individual spatial zone overlooks the spatio-temporal continuity of SLR impacts on a coastal landscape. Such spatio-temporal differences in assessments of populations affected highlight some of the limitations of singular, or limited, spatial zones for adaptation planning.

In a systematic review of research assessing populations affected by SLR, we identify 46 studies that meet our search criteria (See Supplementary Material). Each of these studies assessed populations affected by SLR using varied spatial zones of exposure. The spatial zone of SLR exposure assessed in these studies ranged from mean sea level (narrowest zone) to the LECZ (broadest zone). Twenty studies (43%) used more than one spatial zone, nine (20%) used more than two, and four (9%) used three

or more. The most common three, from narrowest to broadest spatial zone, assessed populations affected as follows: (i) complete inundation or submergence under the future high-tide line (n = 20), (ii) extreme water levels such as storm surge via the 100-year floodplain (n = 17), and (iii) the LECZ (n = 11) (Supplementary Table 1). Of the seven spatial zones used in at least three studies, 61% (n = 28) of the studies used at least one of the three common zones, 13% used two, and only one study[16] used all three. These are the same three common spatial zones identified in previous studies[9,17]. McMichael et al.[17] referred to these three zones as *Specified Levels of SLR*, *Coastal Floodplains*, and the *LECZ* and we adopt those labels here.

Recurrent tidal flooding or flooding on an annual basis and flooding for a specified return level (Coastal Floodplains) is the exposure category where we expect impacts that are most immediate and severe[18–20]. Perigean spring tide events cause regularly recurring water levels well above high tide, and in many coastal areas these cause significant flooding, sometimes referred to as nuisance flooding or recurrent tidal flooding[8], particularly when these tidal events are enhanced by significant onshore winds from tropical cyclones or storms.

Eventually, recurrent tidal flooding gives way to permanent inundation and submergence of coastal areas under future high tides. The future high-tide line (Specified Levels of SLR) is the narrowest delineation of exposure to SLR (i.e. transition from land to ocean), and commonly implies societal impacts that include permanent loss of settled areas, migration, and community relocation[9]. These areas are threatened by inundation and will ultimately be the most adversely affected locations, but societal losses will occur prior *to* permanent inundation and are dependent on adaptive measures that may be undertaken in advance. It is important to note that this strict delineation eschews other hazards associated with SLR.

Areas located beyond the Coastal Floodplains and within the upper bounds of the LECZ broaden populations exposed to coastal flooding and especially its extended impacts while carrying much less of a chance of flooding, but still include associated SLR hazards (such as soil salinization)[5,21]. Simply residing within the LECZ does not guarantee direct exposure to a SLR hazard such as a storm surge, but it does carry increased probability of exposure to the side effects of an extreme event through extended impacts on, for example, livelihood opportunities. For example, some populations residing within the 100-year floodplain may experience recurrent tidal flooding, permanent inundation, storm surges, and saltwater intrusion in coming decades,[8,22–24] while those beyond it are less likely to experience such effects. Broader zones such as the LECZ render any coastal area as "exposed" to SLR in nearly any time period, which makes it difficult to determine exactly *who* is exposed to SLR-related effects and *when* they are exposed. For our study, however, we do not assess these extended impacts, rather we specifically highlight the estimated populations that could be indirectly affected by flooding events triggering other impacts in the LECZ.

There are three main advantages to examining SLR exposure assessments across the three most common spatial zones. First, by accounting for the probability of flooding at locations between the LECZ to the mean higher high water (MHHW) mark into a single analytical framework, we better describe how possible differential impacts vary across space. Different populations within the coastal zone have differential exposure to flooding, allowing for more nuanced discussions of what it means to be "exposed" to SLR.

Second, each individual approach implies varying temporal windows. Specified levels of SLR assume SLR exposure only at the moment when areas are permanently submerged. Conversely, the LECZ represents the most inclusive estimate of exposure to SLR

hazards potentially over millennia with high emissions and future sea level under high emissions will far exceed the LECZ if it is kept fixed relative to present sea level[12]. Between these two extremes lie Coastal Floodplains with a gradient of exposure to multiple hazards associated with the slow, continual rise in water levels.

Third, analyzing the most common approaches provides a framework for examining the entire range of population exposures to SLR impacts. Each spatial zone when applied often assumes homogeneity of impacts by identifying populations exposed based on their presence within each designated zone: people are either inside or outside the LECZ; inside or outside the 100-year floodplain; above or below specified levels of SLR. This equality of exposure *within* each chosen spatial zone ignores the variability within the actual zone itself. Those projected to live under the future high-tide line are exposed to virtually all SLR-associated hazards: soil salinization, recurrent tidal flooding, storm surge, livelihood impacts, shoreline erosion, etc. In contrast, those who live at higher elevations within coastal communities might only be exposed to storm surge and indirect livelihood impacts. Allowing for variation of exposure, as we have done here, permits an examination of multiple scenarios that might unfold along the spectrum of flood exposure in the coastal zone.

While we still use the above approaches and recognize their value in what population exposure assessments indicates related to the extended hazards beyond flooding, taken individually, no one approach to characterizing spatial zone is likely to accurately represent the heterogeneity of hazards associated with SLR nor quantify the spatial zone or timing of exposure to flooding. One approach to this problem is the EAE, a unifying spatio-temporal metric that characterizes exposure across all spatial zones using a finite time period (one year) for planning decisions. Unlike other approaches, the EAE indicates the population exposed to annual flooding by summing the range of annual exposure probabilities over space under any given Representative Concentration Pathway (RCP), an exposure profile that changes over time.

Furthermore, most previous assessments focus on assessing the populations exposed specifically to flooding disregard the annual probability of population exposure to flooding above the high-tide line spanning from the relatively frequent nuisance events (such as a spring/king tide) to 100-year floodplains and beyond. While some studies of SLR impacts have examined expected annual damages (e.g.,[11,25,26]), analysis of the expected annual exposure (EAE) of populations to flooding is relatively new[27–29]. EAE allows for integrating across the most common spatial zones into a single, continuous, model of populations annually exposed to flooding due to SLR from the high-tide line to the 10,000-year floodplain. Few assessments of EAE, however, apply projected estimates of future populations, which could serve as indicators of future impacts. Thus, we combine the EAE model with sub-county population projections in the United States to characterize SLR hazards between 2000 and 2100 under three of the IPCC's Representative Concentration Pathways (RCP 2.6, 4.5, and 8.5) and all five Shared Socioeconomic Pathways.

In this work, based on our own review and previous work[9,17], we analyze population exposure for the three most common spatial zones from the high-tide line to the LECZ. We show how this approach allows for better inter-model comparisons between estimates and, crucially, clarifies their differential exposure estimates related to SLR. Furthermore, we examine the EAE for the same areas and suggest that it benefits adaptation planning by showing the annual increase in populations likely to be directly affected by annual flooding events representing the leading edge of SLR impacts. Although adaptation will occur in the future, we do not account for adaptation measures in this analysis, instead

interpreting potential future population exposure as an indicator of potential impacts. We emphasize that the EAE is not a replacement for the others, which have their own merits, but that it instead standardizes the broad coastal zone range into an all-inclusive spatial region centered on annual flood exposure; a metric that we suggest indicates the rate of change in populations exposed to annual flooding in a manner more easily interpreted for local level adaptation planning.

## Results

**Overall Results**. We find that in the year 2000, the expected number of people in the United States exposed to an annual flood event is just over 600 K people, 150 K people lived below the high-tide line, and 2.4 M people lived in the 100-year flood plain (Fig. 1). The combination of coastal population growth and SLR between 2000 and 2020 has already increased the EAE of the US coastal population by 60% (610 K to 980 K), increased the US coastal population living below the high-tide line by 60% (150 K to 240 K), and increased the US coastal population living in the 100-year floodplain by 45% (2.4 M to 3.5 M), despite just a 25% growth in the entire coastal population over the same period.

As the century progresses, SLR places the US coastal population at increasing chance of exposure to flooding (Fig. 1 and Table 1). Under the SSP2 and RCP 4.5 emission scenarios between 2020 and 2100, we project the EAE to increase 325% to 4.1 M people (2.3–6.4 M); we project the US coastal population below the high-tide line to increase more than 435% to 1.2 million people (0.3–5.1 M); and we project the US coastal population living in the 100-year flood plain to increase 160% to 9.0 million people (3.4 –22.3 M). Lower estimates use SSP3 and the 5th percentile projection in RCP 4.5 while upper estimates use SSP5 and the 95th percentile projection from RCP 4.5 unless otherwise noted. At the same time, we project the population in the 406 coastal counties to increase by just over 40% (133–190 M). Importantly, this indicates that exposure to coastal flood hazards *outpaces* any increased exposure due to coastal population growth.

**Uneven exposure**. Exposure to SLR unfolds unevenly across the US (Fig. 2). In the year 2000, just two counties had over 100 K people in the 100-year floodplain. However, by 2100 and assuming no adaptation, we project nine counties with 100 K people annually exposed to flooding, and 13 counties with 100 K in the 100-year floodplain. In every single county and spatial zone, population exposure increases faster than population growth.

**SLR Metrics Comparison**. For any particular county, it is the combination of the three spatial zones (MHHW, 100-year Floodplain, and LECZ) together that capture the breadth of SLR impacts. Counties with similar exposure profiles along one metric (e.g., MHHW) may have vastly different exposure profiles along the other metrics. For example, despite having 100% of their populations in the LECZ, Currituck County, NC and Orange County, TX exhibit markedly different *total* exposure profiles. Figure 3 shows selected county pairs with one or more similar exposure metrics. Even if three of the four exposure metrics are similar, a fourth metric could still be quite different (e.g., McIntosh County, GA compared to Franklin County, FL).

## Discussion

The three most common spatial zones used to characterize the populations affected by SLR impacts have merits depending on data availability, temporal window analyzed, and amount of anticipated SLR. However, an exposure assessment using only one

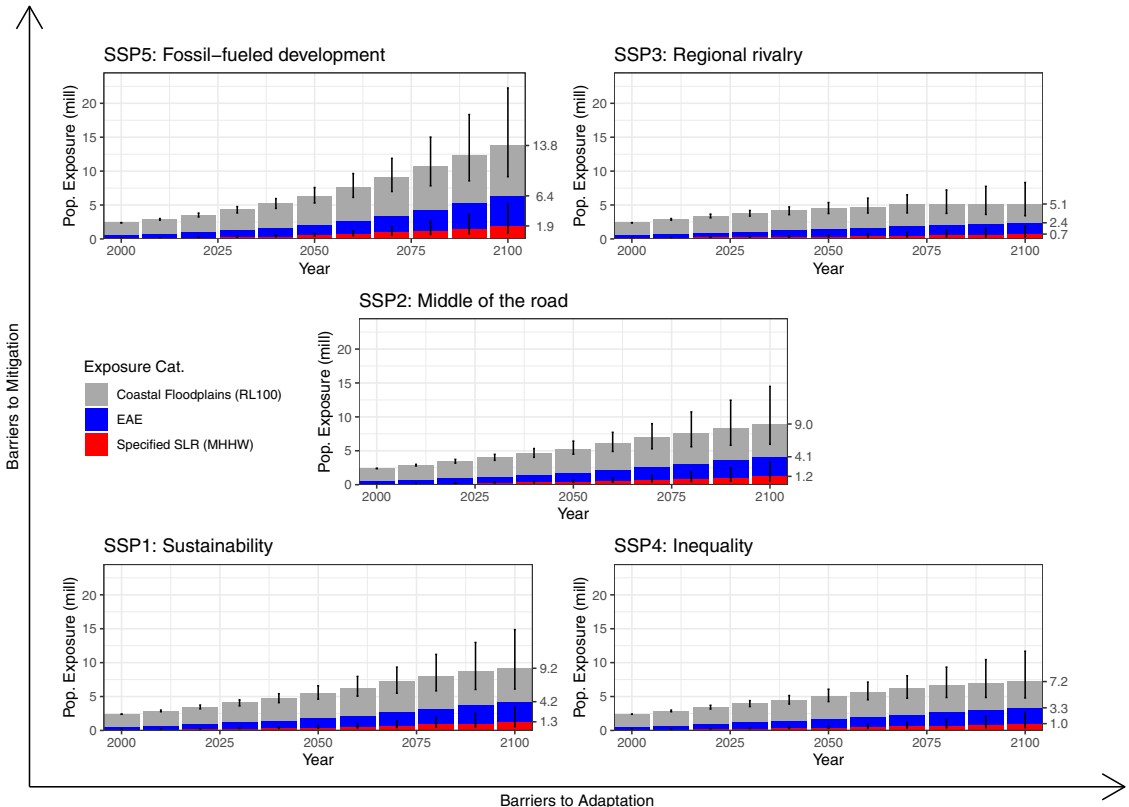

**Fig. 1 Projections of sea level rise flooding exposure under Representative Concentration Pathway 4.5 and all five Shared Socioeconomic Pathways (SSPs) for 2000 to 2100.** Uncertainty reflects the 95th percentile prediction interval. Each spatial zone is not mutually exclusive, but cumulative. 9.2 million people in the Infrequent Flooding Effects zone under SSP1 is inclusive of those in the two preceding zones.

**Table 1 Projected populations affected in all 406 coastal counties exposed across four spatio-temporal zones between 2000 and 2100 under Shared Socioeconomic Pathway 2 (SSP) and Representative Concentration Pathway 4.5 (RCP) in millions.**

| | Specified level of SLR | Coastal Floodplains | | | |
|---|---|---|---|---|---|
| Year | High-Tide Line | EAE | 100-year Flood Plain | LECZ | Total |
| 2000 | 0.15 | 0.61 | 2.39 | 29.00 | 107.15 |
| 2020 | 0.23 (0.2-0.27) | 0.97 | 3.41 (3.11-3.81) | 39.89 (38.92-40.89) | 133.61 (130.84-136.47) |
| 2050 | 0.46 (0.28-0.82) | 1.78 | 5.32 (3.75-7.58) | 52.32 (43.24-62.3) | 165.95 (138.18-195.85) |
| 2070 | 0.74 (0.31-1.78) | 2.63 | 6.94 (3.84-11.88) | 58.79 (42-78.95) | 181.9 (131.42-240.91) |
| 2100 | 1.23 (0.31-5.09) | 4.13 | 8.95 (3.42-22.26) | 63.36 (35.59-100.07) | 190.07 (108.8-293.77) |

Uncertainty intervals in parentheses relate to SSP3, 5th percentile and SSP5, 95th percentile. Total refers to the total population in all 406 coastal counties. Note that each spatial zone and related class are cumulative, not mutually exclusive. For example, the Low-Elevation Coastal Zone (LECZ) estimate encompasses all other spatial zones.

or two of these zones will fail to capture important heterogeneity in SLR exposure and could lead to misguided decision making. We suggest that the lack of consistency across studies and the imprecision of language related to flooding effects may relay a confusing and unclear message to the adaptation planning and policy-making community. For example, MHHW implies that populations below specified levels of SLR will experience imminent property loss and daily flooding. Those inhabiting the Coastal Floodplains zone will experience increasing risk of losses from nuisance flooding, storm surge, saltwater intrusion, soil salinization, etc. (see[9] for review). Populations in the LECZ can prepare for a broader set of socio-economic impacts to livelihoods due to the tangential effects of flooding extending beyond the water line over a large geographic area. For example, extended effects of flooding might include increasing costs of maintaining vulnerable infrastructure, job losses due to declining coastal industries and/or populations[30], the extended effects of coastal

property devaluation[31], or climate gentrification[32]. Our more holistic approach allows for reimagining adaptation planning needs along a continuum of impacts. It is not just who will be affected, but when, and why they will be affected that must be accounted for in adaptation planning scenarios.

Our results indicate that geographic regions that have similar population exposure estimates under one or multiple spatial zones can be different under another zone—sometimes vastly different (Fig. 3). A key insight from our results is a more comprehensive picture for decision-makers who may not otherwise realize which zone in their jurisdiction has the greatest proportional change projected for the population affected by flooding and/or non-flooding related events. Such a misplaced judgement could have significant long-term ramifications for local populations as adaptation strategies may be misguided. For example, Glynn County, GA and Pasquotank County, NC have similar populations exposed to permanent inundation under specified

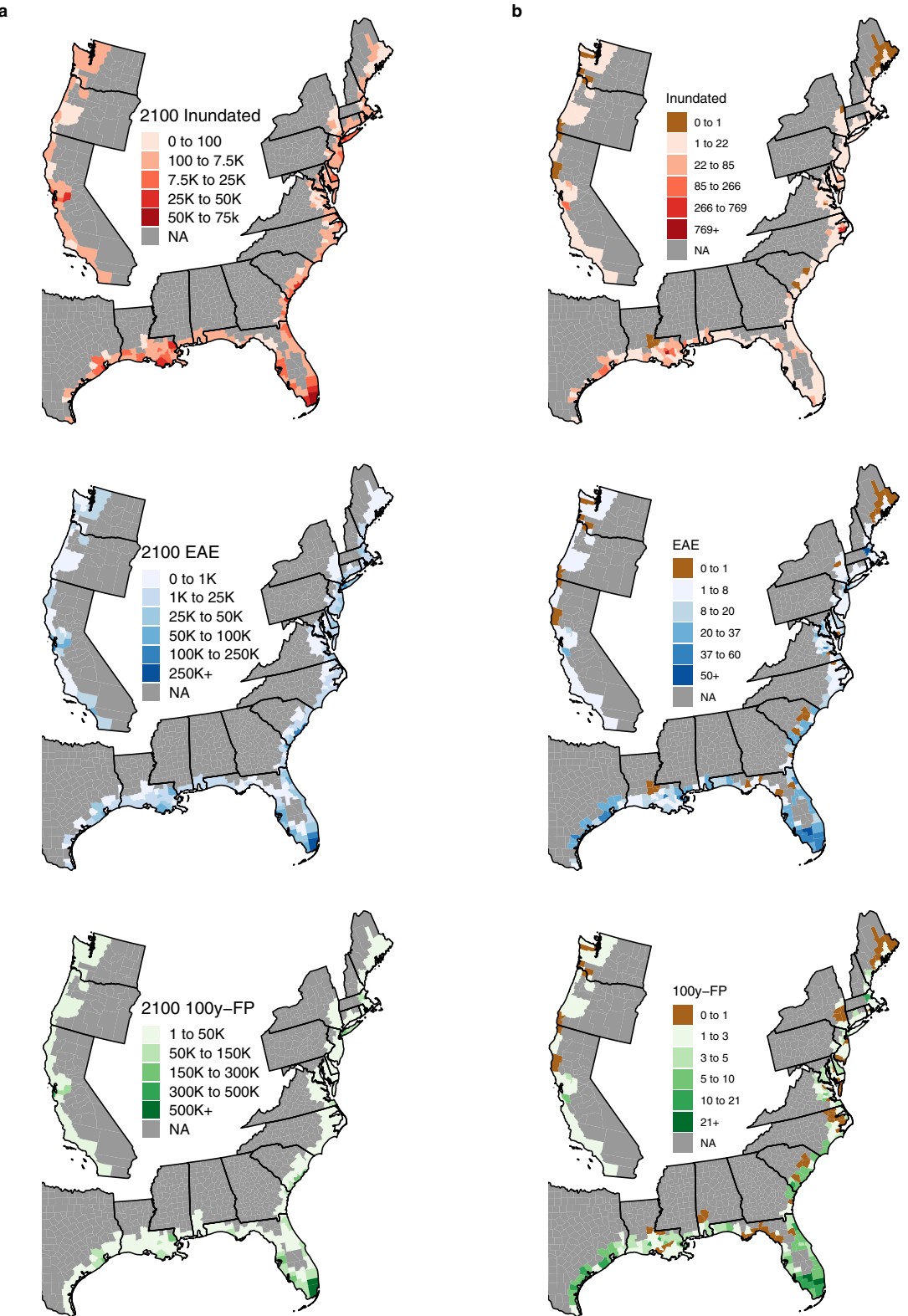

**Fig. 2 Projected populations Inundated under the Mean Higher High Water, Expected Annual Flood, and in the 100-year Flood Plain under Shared Socioeconomic Pathway 2 (SSP) and Representative Concentration Pathway 4.5 (RCP). a** shows the numeric distribution in 2100 and (**b**) shows the relative change in exposure between 2000 and 2100. Brown counties in (**b**) indicates declining exposure.

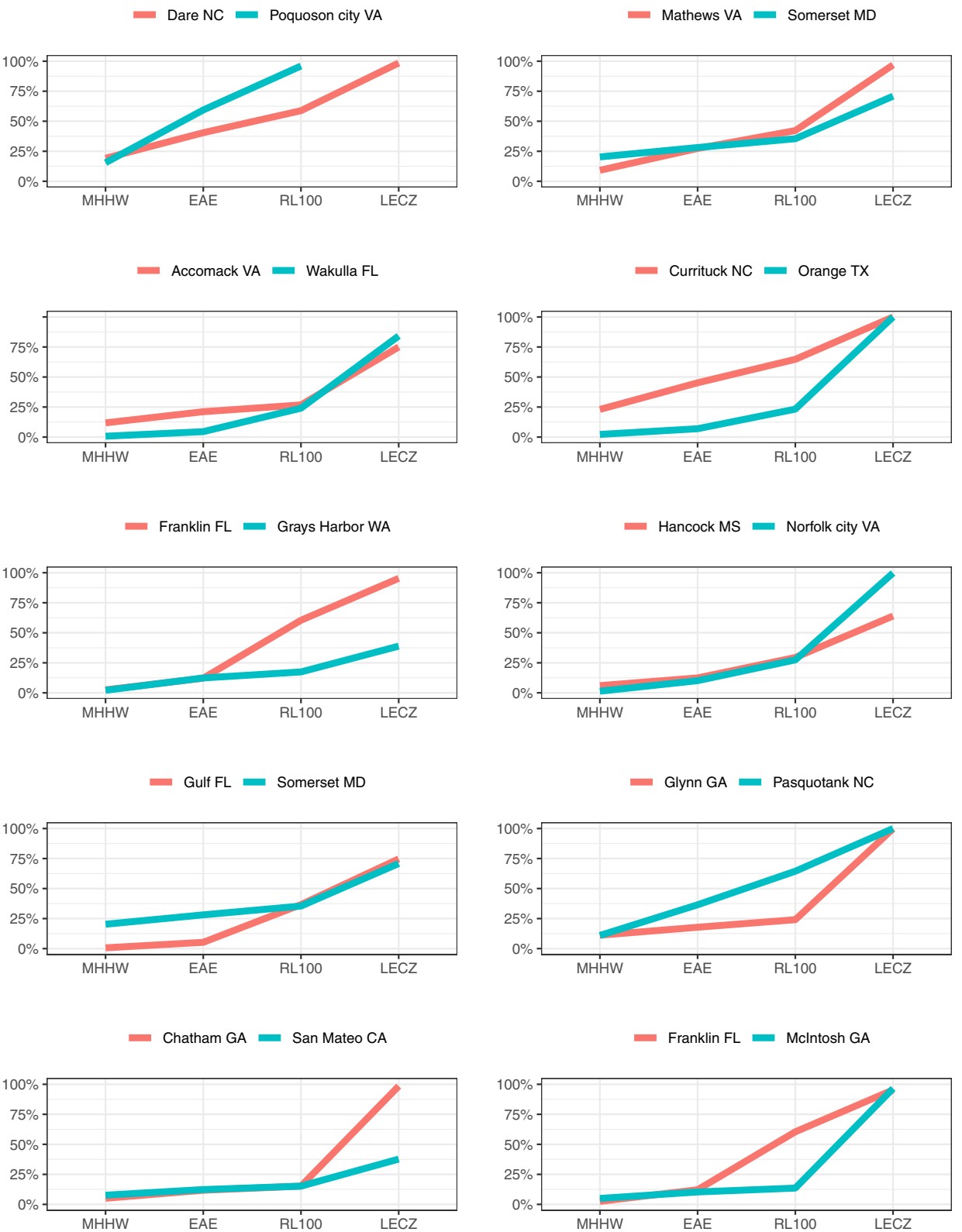

**Fig. 3 The percentage of the projected 2100 population under Shared Socioeconomic Pathway 2 (SSP) and Representative Concentration Pathway 4.5 (RCP) under the four spatial-temporal zones representative of sea-level rise impacts.** Here we compare counties with similar exposures under different spatial zones to show similarity in one zone does not translate to similar exposure under a different spatial zone. MHHW is the Mean Higher High Water, EAE is the expected annual exposure, RL100 is the 100-year Floodplain, and LECZ is the Low-Elevation Coastal Zone.

levels of SLR, and therefore may have similar short-term adaptation responses for those expecting near-term losses (see Fig. 3). However, over the coming century, Pasquotank County, NC has double the population who can expect annual flooding (via EAE) and double who will live in the 100-year floodplain. This suggests that compared to Glynn County, GA, decision-makers in Pasquotank County, NC will need to prioritize long-term adaptation planning and financing, with an eye toward impacts associated with floodplain management over permanent inundation. For federal and state agencies, this type of high-level comparison could inform allocation of national adaptation funding to mitigate the most likely types of impacts aligned with predictions of a region's largest population indicated to be affected. At the local level, it offers decision-makers who have limited resources to allocate toward mitigating impacts the critical details they need to inform tradeoffs in the adaptation planning process, such as the magnitude of populations to be affected, and when, how, and why they will be affected.

Moreover, none of the above spatial zones gives an estimate for the population that will be directly exposed to flooding annually, a useful statistic for planning. The EAE approach's estimate of annual exposure from the 1-year to 10000-year flood plains provides decision-makers with this estimate of the population predicted to be directly affected by a flood event in any given year. Alongside estimates produced from the three most common spatial zones, the EAE allows for both a more nuanced understanding of the exposed population as well as a comparison with other previously published estimates that use the high-tide line, 100-year flood plain, or LECZ in their analysis.

How scientists—and crucially policymakers and planners on the frontlines—conceptualize population exposure to SLR will inform adaptation strategies for mitigating the impacts of SLR. Popular adaptation strategies for SLR include protection, accommodation, and retreat. Typical responses include shoreline armoring, elevating structures, and relocating buildings further from the encroaching shoreline, respectively. Aside from the equity issues of redistributing vulnerability or prioritizing adaptation of privileged populations near the shoreline[33,34], typical adaptation strategies ignore the significant implications to populations and infrastructure located beyond the 100-year floodplain, who will nonetheless experience increasing flooding effects *over time*. Our study highlights the narrowness of popular spatial zones when used alone, but at the same time highlights their conceptual strengths when paired with each other for adaptation planning.

We come to three primary conclusions based on our analysis. Out of our review of 46 articles, scientists studying SLR routinely employ one ($n = 26$) or two ($n = 12$) spatial zones. As we show, restricting a SLR assessment to so few spatial zones overlooks the variation of estimated population exposure across zones, and crucially, does not identify the zones that are experiencing the most rapid change in numbers due to SLR. We suggest that scientists should utilize multiple zones to better quantify SLR impacts for adaptation planning, or at the least be more explicit with their choice of zone and its associated implications. Second, SLR increases anticipated impacts in coastal areas far faster than population growth in coastal areas. Third, using the EAE we show that nearly 1 million coastal residents in the US are presently exposed to an annual flood event. These are SLR impacts presently occurring rather than in the year 2100. Using EAE helps illuminate the contemporary impacts of SLR and its projected annual rate of change, underscoring the importance of immediate adaptation and mitigation needs but also where future efforts could focus on mitigating direct flooding impacts.

While we treat populations as homogeneous groups in our analysis, we recognize that successfully adapting to SLR will require a suite of adaptation planning responses that respond to the range of social and environmental variability within and across coastal systems. Adaptation planning must attend to the ways that social difference affects adaptive capacity under similar exposure levels[33–37]. Achieving equity in adaptation planning[38,39] requires tackling social difference, specifically how differential exposure affects different social groups differently. Moreover, holistic approaches to adaptation planning must also account for the varied range of SLR hazards beyond flooding and the consequent ramifications for socially differentiated populations. Future research might examine social heterogeneity of populations across the most commonly used spatial zones to better capture possible concerns with justice and equity in adaptation planning.

## Methods

**Expected annual exposure**. To demonstrate the effect of spatial zone choice on estimates of populations exposed as an indicator of flooding impacts to SLR for coastal communities, we combined two primary models: a small-area demographic projection model and a flood risk probability model.

**Small-area demographic projection model**. Following[40], we produced a set of small-area demographic projections using a proportional fitting algorithm to produce spatiotemporally consistent Census Block Groups (CBGs) for the period 1940–2010 and employed a mixed, linear/exponential projection for the period 2010–2100. We included only CBGS ($n = 81,815$) located in counties ($n = 406$) expected to experience any probability of flooding.

We produce these projections using the 'Year Structure Built' question, group quarters count ($GQ$), and persons-per household ($PPHU$) from the 2013–2018 Census Bureau's American Community Survey and the count of housing units at the county-level from historic censuses. Following[41], the population in time $t$ in county $i$ in CBG $j$ is given as $P_{tij} = H * PPHU + GQ$.

We calculate $H$ in the period 1940–2010 using $\hat{H}_{ij}^v = \frac{C_j^v}{\sum_{i=1}^n \sum_{t=1939}^{v-1} H_{ij}^v} * \sum_{t=1939}^{v-1} H_{ijt}^v$, where $C_j^v$ is the count of housing units from the historic census in the set of time periods $v \in \{1940, 1950, ..., 2010\}$ in county $j$ and $H_{ijt}^v$ refers to the estimate of housing units in time $t$ from the American Community Survey for block group $i$ in county $j$. For example, to estimate the number of housing units in block group I in county j for the year 1960, the number counted in the 1960 census ($C_j^{1960}$) is divided by the number of HUs in county j as estimated in the ACS for the period 1939–1959 ($\sum_{i=1939}^{1959} H_j^{1960}$) and multiplied by the number of HUs for each block group for the same period ($\sum_{i=1939}^{1959} H_{ij}^{1960}$),

We project $H$ in the time periods 2020–2100 using $H_{ij}^{t+z} = (\alpha + \beta z) + [H^t - (\alpha + \beta t)]$ for any CBGs experiencing population growth and $H_{ij}^{t+z} = e^\beta * z^\alpha + [H^t - (e^\alpha * t^\beta)]$ for CBGs experiencing population decline. We subset our projections for the time periods 2000–2100.

We then control our projections to the Shared Socioeconomic Pathways (SSPs)[42,43]. Out of sample validations suggest reasonably good fit for this approach[40,42]. Controlling our projections to the SSPs allows a near direct translation of our small-area results to national-level SSPs and other nation-level SLR assessments.

**Digital elevation model**. To classify exposure categories, we employed airborne lidar-derived digital elevation models (DEMs) distributed by NOAA[44] supplemented with the USGS Northern Gulf of Mexico Topobathymetric DEM[45] in Louisiana and the USGS National Elevation Dataset[46] in the small fraction of land not covered by the other sources. These elevation data are vertically referenced to NAVD88 and converted to the MHHW datum using NOAA's VDatum grid (version 2.3.5)[47]. Following a bathtub model, we assessed exposed land area using a given water height against the elevation model to generate binary inundation surfaces. The DEM data are high-resolution, high-accuracy, LiDAR-derived digital terrain (bare-earth) models with the lowest uncertainty associated with estimates of flood exposure[11,48,49].

In past literature[28,50,51], it is common to use connected components analysis on binary inundation surfaces to enforce hydrological connectivity to the ocean. While this approach works with a small number of elevation thresholds, it becomes computationally intractable when assessing tens of thousands of SLR scenarios (combinations of years + emissions scenarios + Monte Carlo simulations), as is done in this work. Instead, we follow the framework described in[52] to directly refine the DEMS. First, we generated inundation surfaces from 0–10 m above MHHW, at 0.25 m increments, denoting the $i$'th such height in this sequence by $h_i$, and denoting each such binary surface as ThresholdWaterSurface$_i$(lat, lon). For each pixel in the DEM below 10 m, we noted the minimum value of $i$ for which ThresholdWaterSurface$_i$(lat, lon) is 1 (i.e., where its elevation is below $h_i$), which

we stored in a new index surface *ThresholdIndexSurface(lat,lon)*. We then incorporated levee data from the Mid-term Levee Inventory (FEMA/USACE, acquired September 2013) and used connected components analysis to remove isolated regions within each inundation surface, thus generating fully connected binary masks ConnectedWaterSurface$_i$(lat, lon). As before, for each pixel in the DEM below 10 m, we found the lowest value of $i$ for which ConnectedWaterSurface$_i$(lat, lon), which we again stored in an index surface ConnectedIndexSurface(lat,lon).

We assumed that pixels where ThresholdIndexSurface(lat,lon)= ConnectedIndexSurface (lat,lon) are not isolated, and therefore their elevations in the refined DEM are unchanged. However, locations where ThresholdIndexSurface (lat,lon) < ConnectedIndexSurface'(lat,lon) were isolated. To ensure connectivity when thresholding against new water surfaces, we adjusted such pixels' elevations to equal $h_{ConnectedIndexSurface}$ (lat, lon).

**Sea level rise projections and flood event probability surfaces**. To produce an internally consistent model of flooding, given every pixel in the adjusted DEM, and any SLR projection, we calculated the annual probability that at least one nearby extreme flood event would exceed each pixels' elevation. Here we used the probabilistic SLR projections published previously[53], which incorporate local non-climatic factors such as isostatic adjustment and human-caused land subsidence, and are closely aligned with recent IPCC findings[54,55].

We use historical storm surge records at individual tide stations to estimate their return level curves, and apply them to all pixels between the tide stations using a bathtub model. Unlike studies that perform hydrodynamic simulations on synthetic storms (e.g., FEMA's base flood elevation maps), these curves do not consider factors such as local topography, rainfall, or waves. While the station-distance sensitivity analysis performed in[28] suggests that the distances between tide stations used in this work are sufficiently close to assess EAE in the US, exposure estimates at the <1% probability threshold may be particularly sensitive to these factors.

We specified our model following previous approaches[22,28,52,56] which hold storm surge constant, fitting the parameters of a generalized Pareto distribution (GPD) to historical heights and frequencies of extreme coastal flood events at NOAA tide stations along the US coastline with at least 30 years of hourly records through 2013. This specification allows us to estimate $P(H \geq E|Y = 2000)$, the annual probability of the maximum water height, $H$, exceeding elevation, $E$, in the year 2000 (the baseline year, where SLR=0). We expanded a framework described previously[28,52] to estimate *total* per-pixel annual probability of exceedance of any water height in any year, unconditional to SLR sensitivity to emissions. Published SLR projections[53] are provided as a set of probabilistic distributions, each with 10,000 Monte Carlo samples of SLR for each tide-gauge station and for each year. Below we denote each sample as the function SLR$_j$(y) for $j \in [1, \ldots, 10000]$. From the law of total probability, we can estimate the annual probability of the maximum water height, $H$, exceeds elevation $E$ in year $Y$ from

$$P(H \geq E|Y = y) \approx \frac{1}{10,000} \sum_{j=1}^{10,000} P((H + SLR_j(y)) \geq E|Y = 2000) \quad (1)$$

We computed this function under each emissions pathway (RCPs 2.6, 4.5, and 8.5) for each decade (2000–2100), for elevations between 0 and 10 m at 0.1 m increments. We stored these probabilities in lookup tables for efficient queries.

For every pixel in the DEM with elevation $E(lat,lon)$, we determined its closest NOAA tide-gauge station and used the relevant lookup tables to estimate its annual water height exceedance probability for every SLR projection listed above. We stored the results in a large raster database, producing probability surfaces $P(H \geq E(lat, lon)|Y = y)$ for all three emissions scenarios and decades along the entire US coastline.

Recent studies suggest that the bathtub model employed here likely overestimates exposure, as it does not incorporate wave attenuation nor the time it takes for water to reach their full extent[57,58]. Given the high spatial resolution and wide distributions of water heights used in our EAE analysis, it is not yet computationally feasible to employ a hydrodynamic model to refine these results.

**Exposure computation**. To assess population exposure within a US Census Block Group under any water height (including all exposure approaches described above, namely, MHHW, LECZ, as well as 100-year storm surge adjusted for SLR), we generated a connected inundation surface. For the MHHW and LECZ layers, we simply thresholded the adjusted DEM to find pixels below SLR(y) and for (10+SLR(y)), respectively. For the 100-year storm layer, we thresholded the probability surface to find pixels where $P(H \geq E, |, Y = y) < 0.01$. For each block group, we counted the percentage of its pixels on dry land (as defined by the National Wetland Inventory[59]) covered by the inundation surface, and multiplied by its total population, as predicted by each SSP. To compute expected annual exposure (EAE), defined as the expected number of people on land below the maximum local storm surge height in a given year[28], we multiplied the value of each pixel within the probability surface $P(H \geq E(lat,lon)|Y = y)$ by the block group's (per-pixel) population density and computed the sum.

## Data availability

The data resulting from this study are deposited at https://doi.org/10.5281/zenodo.5562904[60]. The underlying data that support the findings of this study are available from Climate Central but restrictions apply to the availability of these data, which were used under license for the current study, and so are not publicly available. Data are however available from the authors upon reasonable request and with permission of Climate Central.

## Code availability

Code to reproduce our analysis is available at https://doi.org/10.5281/zenodo.5562904[60].

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

## Acknowledgements

This work was supported by the National Socio-Environmental Synthesis Center (SESYNC) under funding received from the US National Science Foundation DBI-1639145 (D.W., V.M., M.H., D.H., S.K., P.C.). We also would like to thank M. Oppenheimer for his insightful comments in developing this paper.

## Author contributions

M.H. and D.H. conceived of the work; M.H., D.H., S.K., V.M., D.W., and P.C. interpreted the results. M.H., D.H., and S.K. wrote the manuscript and M.H., D.H., S.K., V.M., D.W., and P.C. substantively revised the manuscript. M.H. and S.K. created the software.

## Competing interests
The authors declare no competing interests.
