## [Peer Review File · Nature Communications]

REVIEWER COMMENTS

Reviewer #1 (Remarks to the Author):

This paper examines previous sea level rise and flood exposure studies and assesses the 3 most cited categories to spatially assess populations at risk over a range of flood frequencies. I found the methods and results convincing and a useful contribution to the topic. I recommend its publication with only a couple suggested edits (below).

I have one main issue that might be further addressed either in this paper or at least given a bit more discussion with recommendations for further research. What is the spatial uncertainty in the 'closest tide gauge' approach to assigning high water probabilities (MHHW to 100-yr height/LE CZ) that should be factored into any particular census block estimate? Maybe it is a wash? Maybe not. It would be helpful to give at least some insight and/or discussion, e.g., storm-tide return level uncertainty vs. mapping/demographic uncertainty. Also, can the authors please discuss the limitations in using singular tide gauge analysis to also estimate the very rare event (e.g., $\leq 1\%$ annual chance event) at a particular location?

A few other comments:

1) In constructing your digital elevation model, could you compare/contrast to methods of the NOAA SLR Viewer methods since it is heavily used by folks in this field.

2) Since you are using the 100-yr event to delineate an exposure layer, and since this is focused on the U.S., you should at least mention differences from FEMA's BFE definition and note that this estimate is based solely upon tide gauge measurements not inclusive of synthetic storms (connects back to my main issue above) or any other forcing component, e.g., waves, river, direct rainfall not inherent to the tide gauge data.

Reviewer #2 (Remarks to the Author):

The authors have proposed a typology for population exposure assessment that consists of permanently inundated, frequent flooding effects, and infrequent flooding based on three most common spatial zones for coastal population exposure assessment: mean higher high water, the 100-year floodplain, and the low-elevation coastal zone, respectively. They further characterized three zones and estimate the current and future population exposure to coastal flooding under various projections of sea level rise (SLR) and population change. I agree with the authors that a common language for thorough assessment of exposure to coastal flooding is lacking in the community and different characterization of hazard zones makes the comparison hard between estimates. Such unified terminology will have various positive policy implications as well. I also like the way sea level and population in hazard zones are both projected which makes the results more representative of future condition. While the gap is well identified and manuscript is well-written, I am not convinced that the proposed typology is consistent enough to fill in the gap. In short, while annual exceedance probability is a good measure of hazard characterization for less frequent events it simply fails to capture the dynamics of events with frequency more than once a year. And such classification would yield misperception and confusion. So, I recommend a major revision for this submission to give authors an opportunity to revise their proposed typology (if agree with my comments!) and the results and discussions accordingly.

Major comment:

I see a significant inconsistency in the language used here and the common understanding around the flooding severity categories. In this proposed typology, Permanent Inundation, Frequent Flooding Effects, and Infrequent Flooding Effects "represent, respectively, a population's probability of exposure to flooding to be 100%, 1% to 99.9%, and $<1\%$ " (L 123 - 125). While not specified here, my understanding from the remainder of manuscript is that the term "probability" here refers to annual probability. Based on this classification events with return periods from one to 100 years are all considered to be "Frequent" floods (L 323-324). This category is further explained as "Recurrent tidal flooding or flooding on an annual basis (Frequent Flooding Effects) is the exposure category where we

expect impacts that are most immediate and severe”.

To the best of my knowledge, 100yr floods (1% probability) are not considered frequent events and using such terminology for these rare events yields a significant miscommunication. On the other hand, Recurrent tidal flooding (with 99.9% probability; a.k.a. sunny-day flooding or nuisance flooding) refers to low levels of inundation that do not pose significant incident threats to public safety nor cause major property damage (Moftakhari et al., 2018). Thus, “immediate” and “severe” are not good explanatory terms for impacts associated with recurrent tidal floods. In fact the cumulative nature of socio-economic impacts (i.e. traffic and business interruptions) due to these events over a long period of time make these events costly (Moftakhari et al., 2017).

From probabilistic point of view, annual probability fails to correctly describe the dynamics of events repeated throughout the year (i.e. recurrent tidal floods). I mean 100% annual probability of exposure to flooding (which is used here to describe the Permanent Inundation class) simply represents the chance of being exposed to flooding at least once a year, which can be simply a community exposed to tidal flooding few times a year during king tides. But, despite your proposed typology, these areas are not considered permanently inundated and better fit to the definition of frequently flooded areas. Another probabilistic issue here is that, as a rule of thumb, extrapolation of extreme water level events should be limited to return periods no longer than 4x the available record length (Pugh and Woodworth, 2014). Here, the longest record should not exceed 170 years that makes return period estimates less than 600 years reasonable. So, I don't know if results of GDP and Monte Carlo simulations for 10,000 year floods are reliable.

Minor comments:

- L 102-105: The sentence is too long and confusing.
- L 189 – 279: to comply with Nature Communications formatting, you should probably move the methodology to the end of manuscript, after Conclusions.
- L 203: the summation does not seem correct to me. It calculate the sum of Hijt from $t=1939$ till $t-1$? Mathematically it does not make sense! I mean let's start the series with $t=1939$, then what would be the next number and what would be the last number? Please, review it and make sure math is correct.
- L 254: Kopp et al. (2014) provides SLR projections under specific percentiles. Have you interpolated between percentiles to get the full distribution? If yes, please provide the details. Also, seems like references 33 and 37 are identical.
- In the footnote of page 8, it should be 95th percentile?

References

- Kopp, R.E., Horton, R.M., Little, C.M., Mitrovica, J.X., Oppenheimer, M., Rasmussen, D.J., Strauss, B.H., Tebaldi, C., 2014. Probabilistic 21st and 22nd century sea-level projections at a global network of tide-gauge sites. *Earth's Future* 2, 383–406. <https://doi.org/10.1002/2014EF000239>
- Moftakhari, AghaKouchak, A., Sanders, B.F., Allaire, M., Matthew, R.A., 2018. What is Nuisance Flooding? Defining and Monitoring an Emerging Challenge. *Water Resources Research*. <https://doi.org/10.1029/2018WR022828>
- Moftakhari, AghaKouchak, A., Sanders, B.F., Matthew, R.A., 2017. Cumulative hazard: The case of nuisance flooding. *Earth's Future* 5, 214–223. <https://doi.org/10.1002/2016EF000494>
- Pugh, D., Woodworth, P.L., 2014. Sea-level science: understanding tides, surges, tsunamis and mean sea-level changes.

Reviewer #3 (Remarks to the Author):

The manuscript by Hauer et al. develops a typology that classifies different exposure indicators based on a systematic review of broad-scale coastal exposure assessment methods and calculates the expected annual population exposure (called EAE) for the US. The work undertaken is methodologically sound and addresses a topic of scientific significance. However, I have some major concerns about the novelty of the presented results as very similar work and concepts already exist in the peer-reviewed literature. First, a very similar review has recently been undertaken by McMichael et al. (2020) and has been published in *Environmental Research Letters*. The paper reviews 33 publications that provide broad-scale estimates of exposure, classified by (1) population impacted by specific levels of SLR, (2) number of people living in the floodplain of specific return periods and (3) LECZ population (which includes a similar body of literature). Second, the EAE is not a new concept and was introduced in previous studies. For example, Hinkel et al. (2014) and Vafeidis et al. (2019)

have used this concept and have presented estimates on “Expected value of the number of people flooded per year”, a calculation based on elevation and population data and the probability distribution of the hazard (i.e. sea flood heights and their probability of occurrence). Further, Tiggeloven et al. (2020), (Rohmer et al., 2021) or Vousdoukas et al. (2018) used the concept “Expected annual damages” which is also calculated by taking the integral of the exceedance probability curve. Third, it is unclear to me how the developed typology advances research and what the specific benefits are. The authors claim, for instance, that ‘A key insight that our typology provides is a more comprehensive picture for decision-makers who may not otherwise realize which zone has the greatest proportional change projected for the population affected by flooding and/or non-flooding related events’ (l.341-344). However, I am unsure whether there is actually any real advantage compared to using the individual indicators. Different concepts to define exposure are meant to address different questions and different research needs and, in my opinion, they are all valid for specific questions and aims. People living in the annual floodplain are by definition most exposed to a rising sea-level and not all people in the LECZ will necessarily be impacted by a rising sea level (which is clear from the concepts). Therefore, I am currently struggling to see how the presented work is novel. (In case I am missing something that I have not correctly understood from the manuscript I would recommend the authors to clearly highlight the research aim/question/innovation of the manuscript.)

Further comments on study design and methodology:

1. Comment related to the statement in line 60-64:

Exposure analysis considers different sea-level rise and socio-economic scenarios, focusing on different geographic scales and time horizons, and employ different datasets (e.g. elevation, population, extreme sea-level datasets) and methods (e.g. hydrological connectivity, dynamic vs. static approach) which make the comparison of exposure studies in general very challenging/lead to high uncertainties and a huge range of future population exposure.

For instance, the choice of the elevation DEM is a crucial parameter to define/calculate the exposure zones. The error in broad-scale DEMs and the fact that global DEMs are surface models leads to large uncertainty in the estimation of coastal flood exposure. Hinkel et al. (2014) explore the uncertainty and sensitivity of DEMs in broad-scale flood impact assessments. They showed that the 1-100-year coastal floodplain is twice the extent when using GLOBE DEM compared to employing SRTM data. Lichter et al. (2011), analyzed similar patterns by calculating the coastal area that lies below 2m. The authors found that the extent is halved when using SRTM data compared to using GLOBE or GTOPO DEMs. I believe that uncertainties related to the data and methods employed are much more challenging when informing adaptation planning and policies on a broad scale compared to the indicator choice. Therefore, I disagree with the above statement. I think the paper lacks a discussion and (maybe also analysis) on the fact that the underlying data and methods lead to the highest uncertainty in the assessment of current and future exposure to SLR.

2. Is the typology developed for current or future conditions? In line 125 I was wondering if the classes/probabilities are for a specific point in time. Further, the probabilities of the different classes are kind of arbitrary. Have you explored the sensitivity of the different typology classes and assigned probabilities for the EAE calculation?

3. The authors used a static method to calculate coastal flooding in the manuscript (line 219). However, the bathtub method has recently been criticized (mainly due to the fact that it is overestimating flooding). The whole US has recently been modeled using a simplified hydrodynamic approach (LISFLOOD-FP model) by Bates et al. (2021) and I would suggest to use the methods to improve the US estimates in the manuscript.

4. Line 94-96: Several studies have included future population. As I have indicated before, the studies of Hinkel et al. 2014, Vafeidis et al. 2019, Tiggeloven et al. 2020.

5. Other studies that the authors could include in the systematic review: (Muis et al., 2017, Jongman et al., 2012, Kulp and Strauss, 2019, Vafeidis et al., 2019, Mondal and Tatem, 2012, Merkens et al., 2018, Merkens et al., 2016)

6. In case the authors have conducted the analysis using MATLAB/R/Python, it might be helpful to

provide the code. Currently, it is not possible to reproduce the results using the text of the methods section provided in the manuscript.

7. Line 290: The Combination of SSP2 and RCP 8.5 is not plausible as the only SSP that can reach emissions that are high enough to lead to an RCP8.5 forcing is SSP5 (see Rogelj et al. (2018) for more information).

8. Line 178: I find the expression 'multiple scenarios' here a bit misleading as the sentence is related to the different indicators/parameters and not the SLR/Socio-economic scenarios, or?

References:

BATES, P. D., QUINN, N., SAMPSON, C., SMITH, A., WING, O., SOSA, J., SAVAGE, J., OLCESE, G., NEAL, J., SCHUMANN, G., GIUSTARINI, L., COXON, G., PORTER, J. R., AMODEO, M. F., CHU, Z., LEWIS-GRUSS, S., FREEMAN, N. B., HOUSER, T., DELGADO, M., HAMIDI, A., BOLLIGER, I., E. MCCUSKER, K., EMANUEL, K., FERREIRA, C. M., KHALID, A., HAIGH, I. D., COUASON, A., E. KOPP, R., HSIANG, S. & KRAJEWSKI, W. F. 2021. Combined Modeling of US Fluvial, Pluvial, and Coastal Flood Hazard Under Current and Future Climates. *Water Resources Research*, 57, e2020WR028673.

HINKEL, J., LINCKE, D., VAFEIDIS, A. T., PERRETTE, M., NICHOLLS, R. J., TOL, R. S. J., MARZEION, B., FETTWEIS, X., IONESCU, C. & LEVERMANN, A. 2014. Coastal flood damage and adaptation costs under 21st century sea-level rise. *Proceedings of the National Academy of Sciences of the United States of America*, 111, 3292–3297.

JONGMAN, B., WARD, P. J. & AERTS, J. C. J. H. 2012. Global exposure to river and coastal flooding: Long term trends and changes. *Global Environmental Change*, 22, 823–835.

KULP, S. A. & STRAUSS, B. H. 2019. New elevation data triple estimates of global vulnerability to sea-level rise and coastal flooding. *Nature Communications*, 10, 4844.

LICHTER, M., VAFEIDIS, A. T., NICHOLLS, R. J. & KAISER, G. 2011. Exploring Data-Related Uncertainties in Analyses of Land Area and Population in the "Low-Elevation Coastal Zone" (LECZ). *Journal of Coastal Research*, 274, 757–768.

MCMICHAEL, C., DASGUPTA, S., AYEB-KARLSSON, S. & KELMAN, I. 2020. A review of estimating population exposure to sea-level rise and the relevance for migration. *Environmental Research Letters*, 15, 123005.

MERKENS, J.-L., LINCKE, D., HINKEL, J., BROWN, S. & VAFEIDIS, A. T. 2018. Regionalisation of population growth projections in coastal exposure analysis. *Climatic Change*, 151, 413–426.

MERKENS, J.-L., REIMANN, L., HINKEL, J. & VAFEIDIS, A. T. 2016. Gridded population projections for the coastal zone under the Shared Socioeconomic Pathways. *Global and Planetary Change*, 145, 57–66.

MONDAL, P. & TATEM, A. J. 2012. Uncertainties in measuring populations potentially impacted by sea level rise and coastal flooding. *PLoS One*, 7, e48191.

MUIS, S., VERLAAN, M., NICHOLLS, R. J., BROWN, S., HINKEL, J., LINCKE, D., VAFEIDIS, A. T., SCUSSOLINI, P., WINSEMIUS, H. C. & WARD, P. J. 2017. A comparison of two global datasets of extreme sea levels and resulting flood exposure. *Earths Future*, 5, 379–392.

ROGELJ, J., POPP, A., CALVIN, K. V., LUDERER, G., EMMERLING, J., GERNAAT, D., FUJIMORI, S., STREFLER, J., HASEGAWA, T., MARANGONI, G., KREY, V., KRIEGLER, E., RIAHI, K., VAN VUUREN, D. P., DOELMAN, J., DROUET, L., EDMONDS, J., FRICKO, O., HARMSSEN, M., HAVLÍK, P., HUMPENÖDER, F., STEHFEST, E. & TAVONI, M. 2018. Scenarios towards limiting global mean temperature increase below 1.5 °C. *Nature Climate Change*, 8, 325–332.

ROHMER, J., LINCKE, D., HINKEL, J., LE COZANNET, G., LAMBERT, E. & VAFEIDIS, A. T. 2021.

Unravelling the Importance of Uncertainties in Global-Scale Coastal Flood Risk Assessments under Sea Level Rise. *Water*, 13.

TIGGELOVEN, T., DE MOEL, H., WINSEMIUS, H. C., EILANDER, D., ERKENS, G., GEBREMEDHIN, E., DIAZ LOAIZA, A., KUZMA, S., LUO, T., ICELAND, C., BOUWMAN, A., VAN HUIJSTEE, J., LIGTVOET, W. & WARD, P. J. 2020. Global-scale benefit–cost analysis of coastal flood adaptation to different flood risk drivers using structural measures. *Natural Hazards and Earth System Sciences*, 20, 1025-1044.

VAFEIDIS, A. T., SCHUERCH, M., WOLFF, C., SPENCER, T., MERKENS, J. L., HINKEL, J., LINCKE, D., BROWN, S. & NICHOLLS, R. J. 2019. Water-level attenuation in global-scale assessments of exposure to coastal flooding: a sensitivity analysis. *Natural Hazards and Earth System Sciences*, 19, 973-984.

VOUSDOUKAS, M. I., MENTASCHI, L., VOUKOUVALAS, E., BIANCHI, A., DOTTORI, F. & FEYEN, L. 2018. Climatic and socioeconomic controls of future coastal flood risk in Europe. *Nature Climate Change*, 8, 776-780.

Response to Reviewers, Manuscript NCOMMS-21-13175-T

We want to thank the reviewers for their thoughtful feedback and the editors for their guidance. We have documented all changes – point by point – made to the manuscript in the table accompanying this letter.

We also addressed all of R2's and R3's comments regarding more robust support for the typology and applicability. Specifically, we edited all language surrounding the typology to better reflect past work on this topic (as R3 suggested). We've also added additional text and language to better demonstrate our extensions of this past work and emphasized our empirical analysis. We added an additional table outlining some of the confusion in the existing literature that, we believe, bolsters the usefulness of the typology.

R3 requested that we add additional citations to our systematic review. These citations did not meet the qualifications of our search and do not meaningfully alter our results. If we were to add these citations to our review, it would cease to be systematic and adding most of these suggested citations to the manuscript feels forced, since most are only tangentially related (eg, Mondal and Tatem is comparison of gridded population data sets and Muis 2017 is a comparison of different flood zone rasters). Updating our systematic review would add an additional 100 papers to review, many of which continue to use the same spatial zones we identify here (and the same spatial zones McMichael et al 2020 and Hauer et al 2020 also identify). Instead, we've inserted the relevant suggested citations throughout the manuscript. We hope this is a satisfactory compromise.

We believe these changes have substantially improved this paper from its previous form and hope the editors and reviewers find our revisions satisfactory.

Thank you.

Sincerely,

The Authors

Index	Comment	Response
R01-01	I have one main issue that might be furthered addressed either in this paper or at least given a bit more discussion with recommendations for further research. What is the spatial uncertainty in the 'closest tide gauge' approach to assigning high water probabilities (MHHW to 100-yr height/LECZ) that should be factored into any particular census block estimate? Maybe it is a wash? Maybe not. It would be helpful to give at least some insight and/or discussion, e.g., storm-tide return level uncertainty vs. mapping/demographic uncertainty. Also, can the authors please discuss the limitations in using singular tide gauge analysis to also estimate the very rare event (e.g., $\leq 1\%$ annual chance event) at a particular location?	This comment raises an important point regarding the reliability of results between tide stations. Part of our team has already published such a sensitivity analysis in (Kulp and Strauss, 2017), concluding that the vast majority of coastal cities in the US are close enough to a tide station (< 300 km) that EAE and its derived analyses remain valid. We have added a citation to the manuscript to discuss this. We have also added text to the manuscript discussing the limitations of analyses of the 100 year flood event.
R01-02	In constructing your digital elevation model, could you compare/contrast to methods of the NOAA SLR Viewer methods since it is heavily used by folks in this field.	We've edited the second paragraph under the Digital Elevation Models subsection in the Methods to briefly discuss our differences with the NOAA SLR Viewer. In short, the use of binary inundation surfaces (inundated/not inundated) to enforce hydrological connectivity is computationally intractable for our analysis here. Instead, we directly refine the DEMs based on Buchanan et al 2020 .
R01-03	Since you are using the 100-yr event to delineate an exposure layer, and since this is focused on the U.S., you should at least mention differences from FEMA's BFE definition and note that this estimate is based solely upon tide gauge measurements not inclusive of synthetic storms (connects back to my main issue above) or any other forcing component, e.g., waves, river, direct rainfall not inherent to the tide gauge data.	Please see our response to R01-01. We've added text to the manuscript to address this concern.

(continued)

Index	Comment	Response
R02-01	I see a significant inconsistency in the language used here and the common understanding around the flooding severity categories. In this proposed typology, Permanent Inundation, Frequent Flooding Effects, and Infrequent Flooding Effects “represent, respectively, a population’s probability of exposure to flooding to be 100%, 1% to 99.9%, and <1%” (L 123 - 125). While not specified here, my understanding from the remainder of manuscript is that the term “probability” here refers to annual probability. Based on this classification events with return periods from one to 100 years are all considered to be “Frequent” floods (L 323-324). This category is further explained as “Recurrent tidal flooding or flooding on an annual basis (Frequent Flooding Effects) is the exposure category where we expect impacts that are most immediate and severe”.	Thank you for this comment and the opportunity to improve our manuscript. We agree with R02’s comments regarding our typology. We’ve rewritten the Typology section to better align with this comment and comment R03-01, where it’s pointed out that McMichael et al 2020 does a similar review (though lacks an empirical analysis of the spatial zones). Specifically, we’ve adopted McMichael et al’s language and renamed our typology categories accordingly: Specified Levels of SLR, Coastal Floodplains, and LECZ. Our contribution is the application of a SLR typology through an empirical analysis of the United States. We also edited our Typology section accordingly. There should be less confusion and less inconsistency in the language used throughout the manuscript.
R02-02	To the best of my knowledge, 100yr floods (1% probability) are not considered frequent events and using such terminology for these rare events yields a significant miscommunication. On the other hand, Recurrent tidal flooding (with 99.9% probability; a.k.a. sunny-day flooding or nuisance flooding) refers to low levels of inundation that do not pose significant incident threats to public safety nor cause major property damage (Moftakhari et al., 2018). Thus, “immediate” and “severe” are not good explanatory terms for impacts associated with recurrent tidal floods. In fact the cumulative nature of socio-economic impacts (i.e. traffic and business interruptions) due to these events over a long period of time make these events costly (Moftakhari et al., 2017).	See our response to R02-01.
R02-03	From probabilistic point of view, annual probability fails to correctly describe the dynamics of events repeated throughout the year (i.e. recurrent tidal floods). I mean 100% annual probability of exposure to flooding (which is used here to describe the Permanent Inundation class) simply represents the chance of being exposed to flooding at least once a year, which can be simply a community exposed to tidal flooding few times a year during king tides. But, despite your proposed typology, these areas are not considered permanently inundated and better fit to the definition of frequently flooded areas.	See our response to R02-02. But changing “Frequent Flooding” to “Coastal Floodplains” should alleviate this inconsistency and confusion. We’ve also eliminated all language referring to the ‘probability of flooding’ from our descriptions of the categories.

(continued)

Index	Comment	Response
R02-04	L 102-105: The sentence is too long and confusing.	Thank you for this comment. We broke up the sentence for readability. It now reads, " We argue that an applied typology allows for better inter-model comparisons between estimates and, crucially, clarifies their implications for exposure to sea level rise related impacts. Furthermore, the EAE approach benefits adaptation planning by showing the annual increase in populations likely to be directly affected by annual flooding events representing the leading edge of SLR impacts. "
R02-05	L 189 - 279: to comply with Nature Communications formatting, you should probably move the methodology to the end of manuscript, after Conclusions.	We moved our methodology section to the end of the manuscript. Thank you.
R02-06	L 203: the summation does not seem correct to me. It calculate the sum of H_{ijt} from $t=1939$ till $t-1$? Mathematically it does not make sense! I mean let's start the series with $t=1939$, then what would be the next number and what would be the last number? Please, review it and make sure math is correct.	Thank you for this comment. We have revised this formula accordingly and added text to describe the estimation process. The equation should be $\hat{H}_{ij}^v = \left(\frac{C_j^v}{\sum_{i=1}^n \sum_{t=1939}^{v-1} H_{ijt}^v} \right) \cdot \sum_{t=1939}^{v-1} H_{ijt}^v.$ where v is the set of time periods $v \in \{1940, 1950, \dots, 2010\}$. For example, to estimate the number of housing units in block group i in county j for the year 1960, the number counted in county j according to the 1960 census (C_j^{1960}) is divided by the number of HUs in county j, as estimated in the ACS, for the period 1939-1959 ($\sum_{i=1939}^{1959} H_j^{1960}$) and multiplied by the number of HUs estimated in the ACS in block group i for the same period ($\sum_{i=1939}^{1959} H_{ij}^{1960}$). This is repeated for each decade until the most recent time period.
R02-07	L 254: Kopp et al. (2014) provides SLR projections under specific percentiles. Have you interpolated between percentiles to get the full distribution? If yes, please provide the details. Also, seems like references 33 and 37 are identical.	We do not directly use (nor interpolate) the projections under the specific percentiles listed in the (Kopp et al., 2014) manuscript. Instead, we use results from the 10,000 Monte Carlo simulations (for each tide station and each RCP scenario) from that paper to generate the full distributions. This was detailed in the paper under the section "Sea Level Rise Projections and Flood Event Probability Surfaces." We also fixed references 33 and 37. Thank you.
R02-08	In the footnote of page 8, it should be 95th percentile?	Thank you! Corrected

(continued)

Index	Comment	Response
R03-01	First, a very similar review has recently been undertaken by McMichael et al. (2020) and has been published in Environmental Research Letters. The paper reviews 33 publications that provide broad-scale estimates of exposure, classified by (1) population impacted by specific levels of SLR, (2) number of people living in the floodplain of specific return periods and (3) LECZ population (which includes a similar body of literature).	Yes, McMichael 2020 does a similar review as does Hauer et al (2020). Neither use all three common spatial zones to empirically test their similarities/differences. Here, we show how differences in these estimates translate empirically – a key difference and extension.
R03-02	Second, the EAE is not a new concept and was introduced in previous studies. For example, Hinkel et al. (2014) and Vafeidis et al. (2019) have used this concept and have presented estimates on “Expected value of the number of people flooded per year”, a calculation based on elevation and population data and the probability distribution of the hazard (i.e. sea flood heights and their probability of occurrence). Further, Tiggeloven et al. (2020), (Rohmer et al., 2021) or Vousdoukas et al. (2018) used the concept “Expected annual damages” which is also calculated by taking the integral of the exceedance probability curve.	Thank you for the suggestion. We are aware of past work on EAE and, in our paper, do not claim that the concept is new. In the original paper, we cited Hinkel et al 2014, Koks et al 2019, Kulp and Strauss 2017, and Rasmussen et al 2020 which used this metric, and have added (Vafeidis et. al, 2019) in this revision. As we discuss in the paper, the main contribution we add here regarding EAE is that we use this metric to assess projected estimates of future populations. Further, here we assess total probabilities of exceedance through a double integration across the sea level rise and exceedance probability curves. We assess this total annual probability of exposure for every decade and RCP scenario, at every coastal pixel in the US, and then perform a third integration multiplying the per-pixel probability with per-pixel projected population density. To our knowledge, this particular approach has never been used before in the literature.

(continued)

Index	Comment	Response
R03-03	Third, it is unclear to me how the developed typology advances research and what the specific benefits are. The authors claim, for instance, that ‘A key insight that our typology provides is a more comprehensive picture for decision-makers who may not otherwise realize which zone has the greatest proportional change projected for the population affected by flooding and/or non-flooding related.’ However, I am unsure whether there is actually any real advantage compared to using the individual indicators. Different concepts to define exposure are meant to address different questions and different research needs and, in my opinion, they are all valid for specific questions and aims. People living in the annual floodplain are by definition most exposed to a rising sea-level and not all people in the LECZ will necessarily be impacted by a rising sea level (which is clear from the concepts). Therefore, I am currently struggling to see how the presented work is novel. (In case I am missing something that I have not correctly understood from the manuscript I would recommend the authors to clearly highlight the research aim/question/innovation of the manuscript.)	R03 raises a good point and we want to thank R03 for the opportunity to improve our manuscript and expand upon our arguments. The point R03 raises (that different exposure indicators address different questions/research needs) but this point is often lost on both scientists and decision makers. We’ve added language to the Introduction further describing this challenge and added a table to the Introduction showing this confusion. For example, Neumann et al 2015 explicitly link the projected exposure of the LECZ to SLR with displacement in their Abstract. McGrahan et al 2007 also explicitly mention migration in their Abstract as a solution to SLR risk, further emphasize migration away from LECZ in the first paragraph of their Introduction, and ultimately suggest people “be encouraged to move away from the coast” (p.20). Nicholls et al 2011 uses “forced displacement” (in their Abstract) to describe the impact of SLR on the LECZ; Desmet et al 2019 use “will be displaced” (in their Abstract) to describe the impact of MHHW changes; Hinkel et al 2014 use “displace existing people” (p. 3294) in their description of the RL100. So we agree – and even argue R03’s point in the manuscript – that “people living in the annual floodplain are by definition most exposed to rising sea-level and not all people in the LECZ will necessarily be impacted by a rising sea level.” However, while this point feels obvious, it is still bears repeating given the state of the literature and our empirical analysis of an application of a SLR typology is a novel, and useful, contribution to the literature.

(continued)

Index	Comment	Response
R03-04	Comment related to the statement in line 60-64: Exposure analysis considers different sea-level rise and socio-economic scenarios, focusing on different geographic scales and time horizons, and employ different datasets (e.g. elevation, population, extreme sea-level datasets) and methods (e.g. hydrological connectivity, dynamic vs. static approach) which make the comparison of exposure studies in general very challenging/lead to high uncertainties and a huge range of future population exposure. For instance, the choice of the elevation DEM is a crucial parameter to define/calculate the exposure zones. The error in broad-scale DEMs and the fact that global DEMs are surface models leads to large uncertainty in the estimation of coastal flood exposure. Hinkel et al. (2014) explore the uncertainty and sensitivity of DEMs in broad-scale flood impact assessments. They showed that the 1-100-year coastal floodplain is twice the extent when using GLOBE DEM compared to employing SRTM data. Lichter et al. (2011), analyzed similar patterns by calculating the coastal area that lies below 2m. The authors found that the extent is halved when using SRTM data compared to using GLOBE or GTOPO DEMs. I believe that uncertainties related to the data and methods employed are much more challenging when informing adaptation planning and policies on a broad scale compared to the indicator choice. Therefore, I disagree with the above statement. I think the paper lacks a discussion and (maybe also analysis) on the fact that the underlying data and methods lead to the highest uncertainty in the assessment of current and future exposure to SLR.	It is true the choice of elevation model is very important for such analyses. Here, though, we use a high-resolution, high-accuracy lidar-derived digital terrain (bare-earth) model developed by NOAA (RMSE of around 10cm). GLOBE and SRTM are very different – first, these are digital surface models, which are not bare-earth, seeing treetops and building tops as land. These DEMs also suffer from huge error (in many locations, GLOBE contains error approaching 100 meters, while SRTM contains error on the order of 10m). We agree in general with R03’s suggestions and we’ve added the following text to the Methods section: The DEM data are high-resolution, high-accuracy, LiDAR-derived digital terrain (bare-earth) models with the lowest uncertainty associated with estimates of flood exposure (Lichter et al 2010; Hinkel et al 2014; Hooijer & Vernimmen 2021.)
R03-05	Is the typology developed for current or future conditions? In line 125 I was wondering if the classes/probabilities are for a specific point in time. Further, the probabilities of the different classes are kind of arbitrary. Have you explored the sensitivity of the different typology classes and assigned probabilities for the EAE calculation?	We are unsure what R03 means in their first question, but we deploy our typology for past (2000-2015), present (2015-2020), and future (2020-2100) conditions. EAE is an integration under the whole flood probability curve. We also removed the language describing the probabilities of flooding in each category (please see our responses to comments R02-01 and R02-03 where we rename our typology categories to be in line with McMichael et al).

(continued)

Index	Comment	Response
R03-06	The authors used a static method to calculate coastal flooding in the manuscript (line 219). However, the bathtub method has recently been criticized (mainly due to the fact that it is overestimating flooding). The whole US has recently been modeled using a simplified hydrodynamic approach (LISFLOOD?FP model) by Bates et al. (2021) and I would suggest to use the methods to improve the US estimates in the manuscript.	Thank you for the suggestion, however it is not feasible for us to use a hydrodynamic approach in this analysis. First, the resolution of the model used in (Bates et al., 2021) is 30m, compared to the ≤ 5 m horizontal resolution used here, amounting to a 36x difference in pixel density. Second, the EAE analyses do not assess specific water levels, but rather on distributions of flood exceedance probabilities, further multiplying computational costs by orders of magnitude. That said, we have added text to the manuscript noting this limitation.
R03-07	Line 94-96: Several studies have included future population. As I have indicated before, the studies of Hinkel et al. 2014, Vafeidis et al. 2019, Tiggeloven et al. 2020.	Hinkel is already included in our analysis. We softened this language to "Few assessments of EAE..." and added Vafeidis and Tiggeloven.
R03-08	Other studies that the authors could include in the systematic review: (Muis et al., 2017, Jongman et al., 2012, Kulp and Strauss, 2019, Vafeidis et al., 2019, Mondal and Tatem, 2012, Merkens et al., 2018, Merkens et al., 2016)	Both McMichael et al 2020 and Hauer et al 2020 use the same three spatial zones so adding these studies (which didn't qualify for our search at the time) wouldn't change our typology. Instead, we've cited some of these studies at various points throughout the manuscript (Kulp and Strauss and Vafeidis). The other studies are tangentially related and adding them to our manuscript feels forced and adding them to our systematic review would cease to make it systematic.
R03-09	In case the authors have conducted the analysis using MATLAB/R/Python, it might be helpful to provide the code. Currently, it is not possible to reproduce the results using the text of the methods section provided in the manuscript.	Unfortunately, due to licensing restrictions from Climate Central, we are not able to share the code performing the various exposure analyses. However, detailed descriptions of how populations below given elevations are assessed can be found in (Strauss et. al, 2012), (Kulp and Strauss, 2017), and (Kulp and Strauss, 2019). We have added citations to these in the manuscript.
R03-10	Line 290: The Combination of SSP2 and RCP 8.5 is not plausible as the only SSP that can reach emissions that are high enough to lead to an RCP8.5 forcing is SSP5 (see Rogelj et al. (2018) for more information).	Thank you. This is a good point we appreciate you raising. We revised the manuscript to reflect RCP 4.5 throughout.
R03-11	Line 178: I find the expression 'multiple scenarios' here a bit misleading as the sentence is related to the different indicators/parameters and not the SLR/Socio-economic scenarios, or?	Changed to "impacts"

REVIEWER COMMENTS

Reviewer #1 (Remarks to the Author):

The authors have addressed my concerns. Thank you and nice work. Their manuscript is ready for publication.

Reviewer #2 (Remarks to the Author):

Authors have sufficiently addressed my concerns in the previous round. So, I would recommend accept for this submission. Congrats! well done!

Reviewer #3 (Remarks to the Author):

This is the second round of review of the manuscript 'Assessing Population Exposure to Coastal Flooding Due to Sea Level Rise'. The authors have improved the manuscript based on the reviews and replied to most of my comments of the previous round. However, regarding my point on the advantages of the typology compared to using individual exposure indicators (Index: R03-03 in the rebuttal) I am still a bit puzzled. The authors have added some text in the Introduction and included a table discussing the language used in the literature for expressing exposed population that is subject to displacement or migration. I find this not very convincing and possibly not relevant, as I do not see any point in discussing migration here (the indicators are characterizing exposure/risk). Maybe I have not been very clear in my previous comment, but it is still not clear to me how the typology advances research or helps decision-making in the future.

From my point of view, the innovative part of the paper is the approach to calculate the flood event probability surface (which is very "hidden" – this is only clear in the methods part and has not been highlighted in the main part) and the expected annual exposure (EAE) for the US. The work of the flood event probability surface is new and innovative. I find the SLR typology story of the current manuscript a bit misleading and confusing, as it was not clear to me in the first instance what the authors have developed that is new or which gaps the authors are addressing with their work. I see potential in improving the manuscript by changing the story of the work and focusing more on the flood event probability surface and the expected annual exposure (EAE) for the US, and highlighting why the other single indicators (h100, MHW, LECZ) are limited in their suitability for capturing differential exposure across space and time.

Instead of writing that the "SLR typology will better guide adaptation planning and policies..." (Abstract: line34), I would highlight that a flood event probability surface could be better suited for adaptation planning and policies, and point out specifically why and what the benefits are. I believe that it would be more precise and easier for the reader to understand what the approach and actual innovative research of the paper is.

I think the authors could still use the different spatial zones to show the benefits and limitations of such indicators compared to the flood event probability, or how the combination of such approaches could be useful – but I would not call this a typology (which is very misleading/and maybe even wrong).

Table 1: This table is not correct. For instance, Neumann et al. 2015 are not referring to Population that will be possibly displaced when using the LECZ indicator (maybe potentially impacted people, but it does not mean that people have to migrate). I find this table very misleading and would suggest removing it.

Minor comments:

Title: Maybe I would rephrase it to: "Assessing Population Exposure to Coastal Flooding Due to Sea Level Rise and Socio-economic development" (just a suggestion)

Line 47-49: Avoiding adverse consequences also depends on the ability to adapt or respond to such consequences (and one important part are reliable assessments)

Line 62-66: I would argue that a third point is important here: (3) Different datasets and methods to calculate exposure (as they lead to high differences in the number of potentially exposed people)

Line 88: 'small-area population': very vague – this needs a better and more precise wording

Line 94-95: It is not clear from the sentence why the typology allows for better comparisons between estimates.

Line 115: Isn't the LECZ indicator covering a larger exposure area than the 10,000-year floodplain?

Line 123: I would give it a more precise name, 'Coastal Floodplain for specific return periods' or so.

Figure 1: The driver of exposure here is population development. I think that is the main message here, right?

Index Comment		Response
R01-01	The authors have addressed my concerns. Thank you and nice work. Their manuscript is ready for publication.	Thank you!
R02-01	Authors have sufficiently addressed my concerns in the previous round. So, I would recommend accept for this submission. Congrats! well done!	Thank you!
R03-01	This is the second round of review of the manuscript 'Assessing Population Exposure to Coastal Flooding Due to Sea Level Rise'. The authors have improved the manuscript based on the reviews and replied to most of my comments	We removed the migration sentence from the Introduction and the table as suggested by comment R03-05. Please see our response to R03-02 for more details on the typology language,

language used in the literature for expressing exposed population that is subject to displacement or migration. I find this not very convincing and possibly not relevant, as I do not see any point in discussing migration here (the indicators are characterizing exposure/risk). Maybe I have not been very clear in my previous comment, but it is still not clear to me how the typology advances research or helps decision-making in the future.

P03-02 From my point of view, the innovative part of the paper is the approach to calculate the flood event probability surface (which is very “hidden” – this is only clear in the methods part and has not been highlighted in the main part) and the expected annual exposure (EAE) for the US. The work of the flood event probability surface is new and innovative. I find the SLR typology story of the current manuscript a bit misleading and confusing, as it was not clear to me in the first instance what the authors have developed that is new or which gaps the authors are addressing with their work. I see potential in improving the manuscript by changing the story of the work and focusing more on the flood event probability surface and the expected annual exposure (EAE) for the US, and highlighting why the other single indicators (h100, MHW, LECZ) are limited in their suitability for capturing differential exposure across space and time.

We agree that the EAE approach is innovative and novel and have worked to highlight that point more in the intro and discussion sections. However, the EAE is one of two novel contributions to the literature we are presenting here in our manuscript.

We have removed all mention of the “typology” specifically since we agree that we are no longer classifying a range of spatial zones as in our initial submission. But, we are analyzing the three most common spatial zones, our second major contribution, and something that has never done to our knowledge with explicit attention to comparing their different outcomes (Figure 3). We mention that one of the 46 papers we identified in our lit review uses all three, but the intent of their paper is not to highlight the differences in results or that each has a specific role in thinking about adaptation planning for SLR.

McMichael et al’s paper was published in ERL in November 2020, highlighting that our topic of making SLR population exposure assessments more explicit and intentional to inform adaptation planning is at the edge of thinking in this field, but also our (second, but in no particular order) novel contribution advances their work by adding an empirical analysis under the different SSPs for the three most commonly used spatial zones.

We appreciate the reviewer’s comments about emphasizing the EAE and have tried to do that in the intro and conclusions beyond what we’d already done. However, given two other reviewers thought the manuscript publishable, we’re wary of making such a significant shift as claiming the EAE is better than the others, especially since earlier reviews suggested that the three most common spatial zones are useful and have their place in SLR exposure assessments. The EAE is not a catchall replacement, but rather a supplement to these others.

In the Introduction, we edited the following text (emphasis ours. Added text in red):

“In this paper, based on our own review and previous work,^{9,21} we analyze population exposure for the three most common spatial zones from the high tide line to the LECZ. We show how this approach allows for better inter-model comparisons between estimates and, crucially, clarifies their differential exposure estimates related to SLR. Furthermore, we examine the EAE for the same areas and suggest that it benefits adaptation planning by showing the annual increase in populations likely to be directly affected by annual flooding events representing the leading edge of SLR impacts.”

“We emphasize that the EAE is not a replacement for the others, which have their own merits, but that it instead standardizes the broad coastal zone range into an all-inclusive spatial region centered on annual flood exposure; a metric that we suggest indicates the rate of change in populations exposed to annual flooding in a manner more easily interpreted for local level adaptation planning.”

In the Discussion, we edited the following text (emphasis ours. Added text in red):

“Moreover, none of the above spatial zones gives an estimate for the population that will be directly exposed to flooding annually, a useful statistic for planning. The EAE approach’s estimate of annual exposure from the 1-year to 10000-year flood plains provides decision-makers with this estimate of the population predicted to be directly affected by a flood event in any given year.”

P03-03 Instead of writing that the “SLR typology will better guide adaptation planning and policies...” (Abstract: line34), I would highlight that a flood event probability surface could be better suited for adaptation planning and policies, and point out specifically why and what the benefits are. I believe that it would be more precise and easier for the reader to understand what the approach and actual innovative research of the paper is. I

P03-04 think the authors could still use the different spatial zones to show the benefits and limitations of such indicators compared to the flood event probability, or how the combination of such approaches could be useful – but I would not call this a typology (which is very misleading/and may be even wrong).

We have modified this language in the abstract, but more in line with our response to R03-02 above. We are not saying that the EAE is better than the other approaches, but merely a new and novel approach that is useful in its own right of identifying annual exposure estimates. The other three commonly used approaches are useful in their own rights as well, which we articulate in the main text.

Thank you for the suggestion. We have removed all mention of the word typology since we removed the classification aspect of the paper (initial figure 1) after our first round of revisions. A more detailed response is in R03-02.

P03-05	Table 1: This table is not correct. For instance, Neumann et al. 2015 are not referring to Population that will be possibly displaced when using the LECZ indicator (maybe potentially impacted people, but it does not mean that people have to migrate). I find this table very misleading and would suggest removing it.	We removed the table and the addition of the text concerning migration.
P03-06	Title: Maybe I would rephrase it to: "Assessing Population Exposure to Coastal Flooding Due to Sea Level Rise and Socio-economic development" (just a suggestion)	Thank you for the suggestion.
P03-07	Line 47-49: Avoiding adverse consequences also depends on the ability to adapt or respond to such consequences (and one important part are reliable assessments)	Thank you for the suggestion. We added "... the ability to adapt to such consequences. " to the end of the sentence.
P03-08	Line 62-66: I would argue that a third point is important here: (3) Different datasets and methods to calculate exposure (as they lead to high differences in the number of potentially exposed people)	Thank you, we agree. Our paper really addresses the first two considerations (the spatial definitions and temporal horizons). The impact of different datasets and methods is beyond the scope of our paper and introducing a third "gap" in the introduction that we do not fill could be confusing to readers. We went ahead and added a footnote to this sentence that reads: "A third major consideration is the deployment of different datasets and methods to calculate exposure. Examining the contribution of different datasets and methods to wide-ranging estimates of sea-level rise exposure is beyond the scope of this paper." This way we can acknowledge the role of data, datasets, and methods without introducing it as a gap we try to fill.
P03-09	Line 88: 'small-area population': very vague – this needs a better and more precise wording	Thank you for the clarifying suggestion. "Small-area population projections" is technical jargon from Demography and typically describes sub-national areas (in the US context, small-area typically refers to US county and sub-county). For clarity and Nature Communications' broad readership, we changed this to " sub-county population projections. "
P03-10	Line 94-95: It is not clear from the sentence why the typology allows for better comparisons between estimates.	Good point. As this is still the introduction, we modified this sentence to read, "We show how an applied SLR typology..." to better indicate the forthcoming explanation and results.
P03-11	Line 115: Isn't the LECZ indicator covering a larger exposure area than the 10,000-year floodplain?	Thank you for this. Yes, we would agree that the LECZ covers a larger area than 10,000 year floodplain and have amended this sentence accordingly.
P03-12	Line 123: I would give it a more precise name, 'Coastal Floodplain for specific return periods' or so.	This is the terminology that McMichael et al used and which we adopt in our manuscript for consistency. But to be more precise, we added " and flooding for a specified return level " to the paragraph describing the Coastal Floodplains.

P03-13 Figure 1: The driver of exposure here is population development. I think that is the main message here, right?

The inverse is true. Looking at Figure 1 in conjunction with Figure 2 helps. Population is projected to increase by ~40% over the next 80 years but exposure under all three metrics are considerably higher than +50%. We wrote "This indicates that exposure to coastal flood hazards *outpaces* any increased exposure due to coastal population growth" on line 208.

REVIEWERS' COMMENTS

Reviewer #3 (Remarks to the Author):

This is the third round of review of the manuscript 'Assessing Population Exposure to Coastal Flooding Due to Sea Level Rise'. The authors have improved the manuscript based on my previous comments.

Regarding the response to comment P03-08, I am still a bit puzzled as this section does not describe what will be addressed in the paper but is a general statement about which factors lead to high differences in the estimation of exposure. To put this into a footnote is not very convincing from my point of view.

Other than that – I think that the paper is ready for publication. Congratulations!